# Treating Deep-Seated Tumors with Radiodynamic Therapy: Progress and Perspectives

**DOI:** 10.3390/pharmaceutics16091135

**Published:** 2024-08-28

**Authors:** Shengcang Zhu, Siyue Lin, Rongcheng Han

**Affiliations:** 1State Key Laboratory for Structural Chemistry of Unstable and Stable Species, College of Chemistry and Molecular Engineering, Peking University, Beijing 100871, China; zhushengcang@allifetech.com; 2Research and Development Department, Allife Medicine Inc., Beijing 100176, China; 3Department of Biomedical Engineering, Columbia University, New York, NY 10027, USA; sl5470@columbia.edu; 4State Key Laboratory of Molecular Developmental Biology, Institute of Genetics and Developmental Biology, Chinese Academy of Sciences, Beijing 100101, China

**Keywords:** radiodynamic therapy, reactive oxygen species, one-component RDT system, deep-seated tumors

## Abstract

Radiodynamic therapy (RDT), as an emerging cancer treatment method, has attracted attention due to its remarkable therapeutic efficacy using low-dose, high-energy radiation (such as X-rays) and has shown significant potential in cancer treatment. The RDT system typically consists of scintillators and photosensitizers (PSs). Scintillators absorb X-rays and convert them to visible light, activating nearby PSs to generate cytotoxic reactive oxygen species (ROS). Challenges faced by the two-component strategy, including low loading capacity and inefficient energy transfer, hinder its final effectiveness. In addition, the tumor microenvironment (TME) with hypoxia and immunosuppression limits the efficacy of RDTs. Recent advances introduce one-component RDT systems based on nanomaterials with high-Z metal elements, which effectively inhibit deep-seated tumors. These novel RDT systems exhibit immune enhancement and immune memory, potentially eliminating both primary and metastatic tumors. This review comprehensively analyzes recent advances in the rational construction of RDTs, exploring their mechanisms and application in the treatment of deep-seated tumors. Aimed at providing a practical resource for oncology researchers and practitioners, the review offers new perspectives for potential future directions in RDT research.

## 1. Introduction

Cancer has posed a significant threat to human health for decades, prompting the development of various treatment modalities. However, the three conventional methods—surgery, radiotherapy, and chemotherapy—each have their own limitations. Surgeries carry risks of metastasis and recurrence due to large incisions. Additionally, radiotherapy and chemotherapy result in local side effects on surrounding normal tissues under high-dose radiation and systemic side effects with the emergence of drug resistance. The shortcomings of these approaches underscore the urgent need for innovative cancer treatments that can overcome these limitations.

Over a century ago, the term “photodynamic” was introduced, marking the beginning of studies on photodynamic therapy (PDT) [1]. PDT, a non-invasive technique, involves the co-localization of light, oxygen, and photosensitizers (PSs) to produce highly cytotoxic reactive oxygen species (ROS). Compared to traditional tumor treatments, such as surgery or chemotherapy, PDT is more advantageous because of the small wound it causes and its high selectivity and low side-effects. While PDT has shown success in local treatment for superficial cancers, its application to deep-seated or solid tumors is technically challenging due to limitations in light penetration and local irradiation.

To address the restricted penetration depth of visible and near-infrared lights, high-energy rays, such as X-rays and γ-rays, have emerged as promising sources for activating PSs in deep-seated tumors. This approach, known as radiodynamic therapy (RDT), combines low-dose, high-energy radiation with PSs to generate highly toxic ROS for dynamic therapy. RDT presents several advantages, including greater penetration depth, higher excitation efficiency than PDT, fewer side effects than traditional radiotherapy, non-drug resistance, and a non-invasive approach to prevent cancer metastasis and recurrence.

The inception of RDT dates back to 2002, when Shin Hashiguchi and colleagues proposed the use of X-rays to activate the PS acridine orange (AO) for deep-seated musculoskeletal sarcoma [2]. Subsequent studies revealed the potential of utilizing 5-aminolevulinic acid (5-ALA)-induced protoporphyrin IX (PpIX) to enhance RDT effects [3]. However, challenges persisted, including the discrepancy between the energy of rays and the excitation energy of PSs, leading to decreased energy utilization efficiency.

In recent years, nanomaterials containing high-Z atomic number elements have emerged as a solution to enhance the deposition of radiation energy at tumor sites during RDT. Notable developments include scintillating nanoparticles [4], nanoscale metal-organic frameworks (nMOFs) [5,6,7,8], and gold nanoparticles (AuNPs) [9], demonstrating enhanced energy transfer from X-rays to PSs, increased cell destruction efficiency, and notable DNA damage.

This review aims to summarize the progress made in RDT over the past two decades (Figure 1), focusing on therapeutic mechanisms, strategies for improving RDT efficacy through PS design, drug delivery, and microenvironmental control. We also highlight advances in combining RDT with other cancer treatments to enhance specificity and lethality to tumor cells. The discussion also encompasses existing challenges and potential directions for further exploration in the field of RDT.

## 2. RDT and Its Problems

RDT has emerged as a novel cancer treatment method, integrating low-dose, high-energy radiation, such as X-rays, with PSs [20,21,22]. This innovative approach aims to exploit the cytotoxic ROS generated by the interaction of radiation and PSs to induce cancer cell death. The fundamental components of the RDT system include scintillators and PSs. For example, Hashiguchi et al. reported that the AO could be excited by low-dose X-rays to generate singlet oxygen (^1^O_2_), and AO-mediated RDT was effective in eliminating mouse osteosarcoma cells in vitro and in vivo [2]. Kusuzaki and Nakamura et al. then successively applied AO-mediated RDT to clinical studies treating patients with high-grade soft tissue sarcomas [10,11,23]. Though clinical cases reported positive results, such as a lower local recurrence rate and increased specificity due to fluorescence, AO-mediated RDT displayed significant limitations in treating deep-seated or large tumors. Takahashi et al. investigated 5-ALA-mediated RDT on radiation-resistant cancer melanoma and revealed the corresponding molecular mechanism of tumor repression [12]. Lu et al. reported two nMOFs constructed from Hf cluster scintillators and porphyrin-based PS ligands—5,15-di(*p*-benzoato)porphyrin-Hf (DBP-Hf) and 5,10,15,20-tetra(*p*-benzoato) porphyrin-Hf (TBP-Hf)—were structurally optimal for RDT and obtained positive clinical feedback by combining with checkpoint blockade immunotherapy [13]. Subsequently, the Lin group developed a series of Hf-based nanoscale nMOFs and nanoscale metal-organic layers (nMOLs). These materials, with tunable chemical compositions, multifunctionality, extremely high porosity, great biocompatibility, short-term stability, and long-term biodegradability, have been applied to RDT [7,8,13,24,25,26]. In principle, RDT can increase DNA double-strand break (DSB), halt cells at the G2/M phase, enhance ROS generation, and induce immune responses that collectively result in cell cycle disruption [13,14,27,28]. Nevertheless, the low photosensitivity, selectivity, and ROS quantum yield make conventional PS still fall short of the expected therapeutic efficacy [11,12,23].

The scintillator-mediated dual-component strategy utilizes scintillators to absorb and convert X-rays into visible light, which can activate nearby PSs to produce cytotoxic ROS. However, this approach faces significant challenges, including insufficient energy transfer efficiency, low loading capacity, diffusion of PSs, and limited ROS generation rate [15,16,20]. These issues hinder the overall efficacy of the RDT system. Thus, developing PSs that can be directly excited by low-dose X-ray independent of fluorescence resonance energy transfer (FRET) is necessary for high efficacy [15].

## 3. Innovative Single-Component RDT System

To overcome the drawbacks of the dual-component strategy, recent developments have focused on one-component RDT systems based on nanomaterials containing high-Z metal elements. This innovative approach has demonstrated great potential in effectively treating deep-seated tumors, thereby addressing the limitations of loading capacity and energy transfer efficiency.

Zhong et al. firstly designed Ce-doped NaCeF_4_:Gd,Tb scintillator nanoparticles (ScNPs) and proved their X-ray responsive properties [15]. Their proposed mechanism suggests that ROS generation can be enhanced by inhibiting the recombination of electrons and holes of Ce^3+^ ions and Tb^3+^ ions. Herein, electrons in the ground state of Ce^3+^ ions and Tb^3+^ ions can absorb the energy of one secondary electron to an excited state or move toward the conduction band and further react with O_2_, as seen in Figure 2A [15]. Meanwhile, in vivo NaCeF_4_:Gd, Tb ScNPs-PEG demonstrated antitumor efficacy up to 63.67% with the least body weight loss. To maximize ROS production, researchers have focused on improving the reduction reaction of oxygen. BiVO_4_@Bi_2_S_3_ heterojunction nanorods (HNRs) reported enhanced RDT therapeutic efficacy due to the long-lasting separation of electron pairs that can be excited directly by ionizing irradiation [17]. Under both X-ray and NIR irradiation, BiVO_4_@Bi_2_S_3_ HNRs demonstrated extraordinary, complete inhibition of tumor growth with a relatively low body weight loss. The heterojunction structure enables electron migration between the conduction band and valence band of BiVO_4_@Bi_2_S_3_. Combined with oxygen vacancies on the surface of PSs, the electron-pair lifetimes are prolonged, thereby increasing the reduction of O_2_ to form ROS, as seen in Figure 2B [17]. 

More recently, gold nanomaterials have been investigated for applications in PDT as PSs [29,30,31] and in radiotherapy as radiosensitizers [32,33,34,35,36,37] owing to their stronger X-ray absorption properties, as gold is a high-Z element. Han et al. proved that dihydrolipoic acid-coated gold nanoclusters (AuNC@DHLA) obtain two-photon excitation properties and enhance type I energy transfer, which produces more responsive ROS free radicals for PDT [29]. After irradiation, cells demonstrate apoptotic morphological changes. Meanwhile, the excess ROS releases lysosomal proteases and damages mitochondria. Gold nanoclusters utilizing the type I photochemical mechanism have proven to be the most efficient in reducing tumor sizes among conventional and type II PSs, while also exhibiting the least toxicity. Therefore, they can be ideal substitutes for PSs in RDT in the future. Recently, AuNCs@DHLA RDT without additional scintillator or PS assistance has been developed [18]. It has been proven to be highly efficient in the in vivo treatment of solid tumors via a single dose of drug administration and low-dose X-ray radiation. Interestingly, enhanced antitumor immune responses have also been observed, which could be effective against tumor recurrence or metastasis (Figure 2C–E).

## 4. RDT Mechanisms

### 4.1. ROS Generation in RDT

RDT primarily relies on the generation of cytotoxic ROS to induce damage to tumor cells. Singlet oxygen (^1^O_2_), superoxide anion radical (O_2_^•−^), hydroxyl radical (OH^•^), and hydrogen peroxide (H_2_O_2_) are the ROS species generally involved [38,39,40,41,42,43]. Based on the mechanisms of ROS generation, RDT can be classified into two types: type I and type II. Type II RDT generates highly reactive ^1^O_2_ through energy transfer from activated PSs to the triplet ground molecular oxygen (^3^O_2_) [44]. In contrast, type I RDT produces ROS via electron (or hole) transfer from PSs to O_2_ (Figure 3). Currently, RDT systems predominantly rely on type II photochemical reactions, which are heavily dependent on and consume large amounts of O_2_ during treatment. However, the hypoxic microenvironment prevalent in deep-seated tumors poses a significant challenge. This issue has led to the exploration of type I RDT, where the PS and substrate molecules undergo direct electron transfer, typically forming free radicals. Importantly, this photodynamic pathway operates without requiring significant oxygen consumption.

### 4.2. The Biomedical Mechanism behind RDT

The precise biomedical mechanism of RDT is still not fully understood, yet the synergistic effect of PDT and hypofractionated radiotherapy is commonly associated with RDT [45]. The following five aspects are known for the biomedical mechanism behind RDT.

#### 4.2.1. The Cytotoxic Effect of ROS

In their study of AO-mediated RDT on mouse osteosarcoma, Hashiguchi et al. reported that the inhibitor of ^1^O_2_, L-histidine, did not inhibit the survival rate of mouse osteosarcoma cells [2]. Thus, they primarily proposed that the generation of ^1^O_2_ might explain the cytocidal effect. Fan and his group worked on the synthesis of hollow mesoporous organosilica nanoparticles (HMONs), a radiosensitizer that hybridizes the organic/inorganic framework, to co-deliver tert-butyl hydroperoxide (TBHP) and iron pentacarbonyl (Fe(CO)_5_) [19]. They found that X-rays can break the O-OH bond of HMOP-TBHP/Fe(CO)_5_ and form highly cytotoxic OH^•^ that brings oxidative damage to DNA and other biological macromolecules in RDT. Studies on nanoscale metal-organic frameworks and nanoscale metal-organic layers showed that the interaction between these materials and X-ray irradiation generate intracellular OH^•^, ^1^O_2_, and O_2_^•−^ during radiotherapy–radiodynamic therapy (RT-RDT) [7,13,26]. Takahashi, Hasegawa, and their colleagues proved that 5-ALA/protoporphyrin IX-mediated RDT promoted the production of free oxygen radicals [3,12,14,27,28]. Meanwhile, the increase of ROS in cells enhances the expression of caspase-3 and caspase-9 (enzymes initiating proteolytic cleavage) in downstream apoptotic pathways, resulting in dysfunction of mitochondria and apoptosis [7]. Though the enhancement of ROS production in RDT has been proved, evaluating the specific contribution of each type of ROS in killing tumors can be challenging due to the characteristics of ROS, including short lifetime, high reactivity, diffusion, and interconversion [12,38,43,46]. Comprehensively, ROS generation destructs DNA double strands, proteins, the panniculus adiposus of organelles (i.e., mitochondria) of tumor cells, and other active biological macromolecules to suppress cell proliferation and prevent cell reproduction [3,14].

#### 4.2.2. The Disturbance of Cell Cycle

Apart from ROS generation, the inhibition of cell mitosis and the disturbance of cell cycle also play an important role in RDT. Kusuzaki and Takahashi et al. reported that aberrant nuclear morphologies and swollen cells were observed after RDT [2,10]. DNA ploidy analysis showed that many of the surviving cancer cells were arrested at the G2 phase and had a polyploid DNA content greater than octoploid [2,10]. Additionally, taking in the ROS scavenger vitamin C can prevent B16-BL6 melanoma cells from being arrested at the G2/M phase [12]. Those vitamin C-treated tumor cells can proceed with the normal cell cycle of growth and cell division, indicating that ROS may disrupt the cell cycle [12]. Many researches also revealed that the oxidative damage of RDT resulted in the DNA double-strand break (DSB, known as the most deleterious type of DNA damage) of tumor cells [7,13,15,19,26,36,47,48]. The abnormality of the DNA prevents cells from passing the G2 checkpoint to enter the M phase, proving that RDT is capable of inhibiting mitosis. 

The cell cycle is precisely regulated by the interactions between cyclins (Ccn), cyclin-dependent kinases (CDKs), and cyclin-dependent kinase inhibitors in eukaryotic cells [48]. Takahashi et al. reported that RDT could decrease the gene expression of *Ccna1*, *Ccna2*, *Ccnb1*, *Ccnb2*, *Cdk1*, and *Cdk4* while up-regulating that of *Gadd45a* and *Cdkn1a* (p21) [12]. CDKs are primarily regulated by binding to their cyclin partners and CDK inhibitors [49]. Since cyclin B activates CDK1 to initiate mitosis [50,51], a decrease of cyclins and Cdk1 indicates cell arrest at the G2 phase. GADD45 is a protein induced by DNA damage or captured by cell cycle, while p21 is an inhibitor of cell cycle-dependent kinase [12]. Thus, the up-regulation of *Gadd45a* and *Cdkn1a* (p21) prevents the cells from progressing the cell cycle at the G2/M checkpoint. 

Malignant tumors are always considered life-threatening because of their uncontrolled growth of abnormal cells. Once damaged cells enter the cell cycle, they can proliferate uncontrollably due to the failure of checkpoints. RDT that can induce the up- or down-expression of genes regulating cell cycle is prospective as a cancer therapy for compensating the unfunctional checkpoints. Meanwhile, the abnormality of DNA renders those cells incapable of passing the functional checkpoint, thereby reducing the aggregation of damaged cells.

#### 4.2.3. The Effect of Directly Killing Tumor Cells

Apoptosis, necrosis, autophagy, and other cell death can serve as regulatory mechanisms of RDT. Apoptosis is an autonomous cell death which is strictly regulated by multiple genes that are highly conservative among species, such as the *Bcl-2* family, *caspase* family, and *p53* [51,52,53,54]. Following RDT’s application, depolarization of the mitochondrial membrane potential and the release of cytochrome c from mitochondria were observed (Figure 4) [7]. Meanwhile, the expression of apoptosis-related proteins caspase 3, caspase 9, and p53 were distinctly up-regulated [7,19,47,55]. Apoptotic cells initially show nucleus and cytoplasm condensation and DNA fragmentation followed by blebbing of the plasma membrane to form apoptotic bodies, which can rapidly be identified by neighboring phagocytes without the induction of inflammation [52,53,56].

Compared to immune-silent cell apoptosis, necrosis is an uncontrolled type of cell death. It begins with cellular swelling, membrane integrity rupture, and spillage of cell contents into surrounding tissues. Eventually, it leads to activation of the immune system and extensive inflammation [57,58]. As early as 2002, Hashiguchi reported that cellular morphological changes such as cytoplasmic swelling, nuclear swelling, and cell membrane rupture were observed after receiving AO-mediated RDT [2]. This suggests the high likelihood that RDT was able to induce necrosis in tumor cells. 

Autophagy involves the engulfment of cellular components such as macromolecules or entire organelles by double-membrane vesicles (autophagosomes). Subsequently, these autophagosomes fuse with lysosome to degrade the enclosed substances [57]. As expected, autophagy also contributed to the killing effect of RDT, as demonstrated by the cell death of B16G4F cells induced by 5-ALA-mediated RDT [28].

Apoptosis and necrosis are merely induced under X-irradiation, while the effects vary across different structures and irradiation doses. Nanoscale MOF-mediated RDT can enhance RT and induce RDT to increase the rate of apoptosis and necrosis of tumor cells. Under 2 Gy X-ray irradiation, DBP-Hf induces 31.5% apoptosis and 1.73% necrosis [13]. Hf12-Ir can induce apoptosis up to 38.6% and necrosis up to 15.1% [26], while Hf-DBB-Ru leads to 62.8% apoptosis and 3.68% necrosis [7]. HMOP-TBHP/Fe(CO)_5_ caused 42.18% apoptosis and 0.65% necrosis under 6 Gy irradiation [19]. 

Apoptosis and necrosis have been the primary types of tumor cell death reported in RDT [7,13,19,26], while other forms of cell death, such as pyroptosis [59], necroptosis [57,58], and ferroptosis [60,61], have received limited attention. Accordingly, further research is needed to investigate different cell death mechanisms.

#### 4.2.4. Activation of the Antitumor Immune Response

X-ray irradiation can activate the immune system and provoke both innate and systematic immune responses. As shown in Figure 5, Ni reported the immuno-oncology cycle and the use of immunotherapeutics to enhance essential steps of immune responses [6]. Immunogenicity calreticulin (CRT) is a chaperone protein, and high-mobility group box-1 (HMGB-1) is a nonhistone chromatin-associated protein. Both are markers of the immunogenic cell death (ICD) as they elicit immune responses from anti-tumor macrophages and dendritic cells. The Lin group discovered that DBP-Hf-mediated RDT can result in the ICD of CT26 cells by the eversion of CRT on the plasma membrane [8]. They also found that ICD can be induced by Hf-MOL (nanoscale metal-organic layer) combined with X-ray treatment, as evidenced by the detection of cell-surface exposure of CRT, HMGB-1 release, and adenosine triphosphate excretion [62]. Takahashi et al. found that RDT led to the up-regulation of representative genes of immune response, MHC class I, and death receptors such as *Fas*, *Icam*, interleukin *IL6*, interleukin *IL15*, Toll-like receptor *Tlr6*, and Toll-like receptor *Tlr7* [12]. The RDT-mediated tumor cell death not only targets cells within the irradiation field but also induces systematic immune responses from metastatic tumors [18]. The DBP-Hf binding to an interference molecule that can reverse immunosuppression were proven to effectively reduce the distant metastatic tumor size [13]. Additionally, Formenti and his colleagues reported that irradiating normal tissues of a mouse’s leg did not cause the abscopal effect. Irradiation on normal tissues could not induce inhibitory regulation on metastasis in non-radiation sites, indicating that anti-tumor systematic immune response can only be evoked after tumor cell death or other stress responses [63]. 

From 1969 to 2014, 46 cases of abscopal response caused by radiotherapy were reported [64,65]. The innate immune response non-specifically induced cell death to adjacent cells through destabilizing the genome and mutating genes, while the abscopal effect on metastasis is anti- or pro-tumor and is synergistically determined by the radiation dose and tissue type. The corresponding signal pathway of RDT-induced systematic immune response has not been clearly investigated yet, but it is anticipated to not only eliminate primary tumors but also prevent their metastasis and recurrence after manipulation.

#### 4.2.5. Advances in Overcoming TME Barriers

The TME is uniquely distinct from that in normal tissues. It is often characterized by the asymmetric distribution of nutrients, low levels of oxygen (hypoxia), acidic pH (acidosis), high levels of reductant (glutathione), and elevated H_2_O_2_ concentration [66]. Substantial limitations on RDT efficacy are primarily posed by the TME. Recent breakthroughs aimed at overcoming these obstacles and enhancing RDT efficacy are discussed.

The oxygen deficiency in tumors reduces radiation-induced DNA damages through the reduction of DNA radicals and promotes radiation resistance by up-regulating the expression of hypoxia-inducible factor 1α (HIF-1α) [67]. HIF-1α that up-regulates genes controlling angiogenesis, metabolism, and metastasis is an important indicator of poor prognosis after radiotherapy [68]. As the cellular oxygen level in a solid tumor determines the extent of cancer cell damage during radiation, it is necessary to develop effective strategies to alleviate tumor hypoxia to promote RDT. Tumor hypoxia is primarily caused by vascular abnormalities that fail to transport O_2_ to cells. Therefore, artificial oxygen carriers are designed to replace red blood cells to address the insufficient oxygen delivery. Yu et al. reported that PFC (a carbon–fluorine compound) emulsions developed to combine with carbogen (95% O_2_ + 5% CO_2_) showed distinct radiosensitization that enhances oxygen concentration in the TME [69]. The sensitization was limited in the tumor hypoxic cells, minimizing cytotoxicity to the surrounding normal cells with appropriate oxygen levels [67]. Beyond increasing the amount of O_2_ transported to the tumor site, drugs have been designed to actively generate oxygen. Ni et al. reported a biomimetic Hf-DBP-Fe nMOF that can decompose elevated concentrations of hydrogen peroxide in hypoxic tumors to produce oxygen and hydroxyl radicals to optimize RDT effects [16]. Apart from degrading H_2_O_2_ in situ to generate oxygen, Fan et al. developed a biodegradable thioether-hybridized HMON to deliver tert-butyl hydroperoxide (TBHP), where the peroxy bond could be cleaved to generate OH^•^ for RDT [19]. This strategy is distinctly effective under hypoxia in TME since the generation of OH^•^ is oxygen independent. In addition, Sang et al. discovered that Hb@Hf-Ce_6_ NPs might take a dual responsibility of increasing the absorbance of radiation energy and boosting the generation of ROS from the Hf-polyphenol coordination platform [70].

The hypoxic microenvironment of deep-seated tumors poses challenges for type II RDT. Although the mechanisms of type I are still debated, numerous studies have demonstrated that type I PDT remains effective under low O_2_ levels. This suggests that type I PDT can be a basis for innovative solutions to overcome hypoxia limitations in type II PDT [71,72,73,74,75,76]. 

Recently, a highly efficient RDT system with type I mechanism was developed solely based on gold nanoclusters (AuNC@DHLA) [18]. The mechanism behind the efficient type I process likely involves the complex energy level structure of AuNC and the ultra-long triplet state. Upon X-ray irradiation, the excited singlet state (^1^AuNC@DHLA*) is generated, which then undergoes efficient intersystem crossing (ISC) to an excited triplet state (^3^AuNC@DHLA*). In other words, the triplet formation originates from an effective excited-state relaxation, where the initially populated singlet (S1) forms triplet (T1) states via an intermediate triplet (T2) state. The low reduction potential and ultralong lifetime of the T1 state facilitate the efficient generation of O_2_^−•^ through intermolecular charge transfer to molecular oxygen. Additionally, the energy gap of T1-S0 is smaller than that between ^3^O_2_ and ^1^O_2_, thereby precluding the generation of ^1^O_2_ by the type-II process [77]. Meanwhile, previous studies have proved that thiol compounds could generate superoxide via an autoxidation reaction [78,79]. Thus, excess thiols or free thiols liberated from the Au surface may also participate in oxidations catalyzed by thiol-protected AuNCs to generate ROS.

Tumor cells survive and thrive in highly acidic environments by controlling their intracellular pH. While the acidic extracellular pH facilitates the invasion of tumors and inhibits immune surveillance, the neutralization of acidity can reverse it to inhibit tumor cell growth [80]. Prasad et al. discovered that MnO_2_ nanoparticles (NPs) can react with endogenous H_2_O_2_ and H^+^ to produce significant amounts of oxygen (~6-fold increase) in situ under radiation, increasing the intracellular pH and producing ROS [81]. Similarly, MnO_2_ conjugated with albumin was proven to relieve tumor hypoxia and low pH for a prolonged time. This system down-regulates HIF-1α and its downstream proteins involved in tumor cell metabolism, angiogenesis, and metastasis [68,81].

Characterized by aberrant oxygen concentration, nutrient distribution, and acidity, TME impedes normal cell division and regulation but boosts tumor cell growth and evasion towards its surrounding areas. It would be important to disrupt the complex interactions between tumor cells and their microenvironment as an effective cancer therapy.

## 5. Combination Therapy

The ideal cancer therapy is expected to both eradicate primary tumors and prevent their metastasis and recurrence. In recent years, RDT has developed to combine with other cancer therapies, such as radiotherapy, checkpoint blockade immunotherapy, gas therapy, and photothermal therapy (PTT), for enhanced therapeutic effects.

Radiotherapy utilizes high-energy photons to ionize radiosensitizers, inducing Compton scattering and photoelectric emission, and then releases excess energy as Auger electrons (short-range secondary electrons) [20]. The Compton and emissive electrons directly damage double-strand DNA, while the Auger electrons transfer the absorbed energy to surrounding water and oxygen molecules, generating ROS to damage biomacromolecules. By employing high-energy radiation rays to excite PSs, RDT enhances both PDT and radiotherapy as those PSs can absorb radiation energies as radiosensitizers. For example, NaCeF_4_:Gd, Tb ScNPs, and POM nanoclusters mitigate the excitation energy gap by internal energy transfer, while their intrinsic absorption of X-rays enables them to produce Auger electrons as well [15,82]. The Hf-based nMOFs and nMOLs are also proven to amplify the RT effect by producing hydroxyl radicals [7,26]. In this way, the proper PSs can take advantage of RT-RDT upon X-ray irradiation. 

Immune checkpoint blockade (ICB) inhibits immune checkpoint pathways cytotoxic T-lymphocyte protein 4 (CTLA4) and programmed cell death protein 1 (PD-1), which negatively regulate T cells, to reinvigorate antitumor immunity [64]. However, the low exposure to the antigen of patients leads to limited immunotherapy responses [6,83]. In recent years, the Lin laboratory has reported that nMOF-mediated RDT synergizes with checkpoint blockade immunotherapy and can eradicate metastatic and recurrent tumors [6]. In 2018, they firstly combined Hf-DBP-mediated RDT with immune checkpoint molecule indoleamine 2,3-dioxygenase (IDO) [13]. As shown in Figure 6A, IDO inhibitors reject the IDO-catalyzed oxidation of tryptophan and boost the production of kynurenine to reactivate T cells in the TME. Hf secondary building units absorbed X-ray irradiation energy and transferred energy to the anthracene ligand DBP to generate ^1^O_2_ in vitro and in cells by the RDT mechanism. Intratumorally injected IDOi@nMOFs treated with low doses of X-ray irradiation were proven to eradicate primary tumors and distal tumors in mouse models of breast and colorectal cancer. Soon afterward, Ni et al. reported that nMOF-mediated RDT combined with an anti-PD-L1 antibody significantly diminished in situ and metastatic tumors [62]. PD-L1 blockade reversed T cell exhaustion to promote the increase in the ratio of CD8^+^ T cells to immunosuppressive Treg cells and T cell oligoclonales to overcome immunosuppression [84]. In addition to targeting adaptive immune checkpoints for multiple metastatic and treatment-refractory cancers, recent studies have shown that innate immune checkpoint CD47 is critical for cancer cells to escape from immune surveillance such as macrophage attack and phagocytes [85,86]. Sang reported that an oxygen-enriched X-ray nanoprocessor Hb@Hf-Ce6 nanoparticle was developed for improving the therapeutic effect of RT-RDT, enhancing the modulation of hypoxia TME and promoting anti-tumor immune response in combination with PD-1 immune checkpoint blockade, as shown in Figure 6B [70]. The Lin group reported that surface modification of Hf-DBP reactivates innate immunity by co-delivering a Toll-like receptor 7 (TLR-7) agonist, imiquimod, and an anti-CD47 antibody (αCD47) in Figure 6C. TLR-7 agonists repolarize immunosuppressive M2 phenotype to immunostimulatory M1 phenotype and promote tumor cell phagocytosis. Under X-ray irradiation, IMD@Hf-DBP/αCD47 effectively activated innate immunity, and IMD@Hf-DBP/αCD47(+) combined with αPDL1 orchestrated adaptive immunity that completely eradicated both primary and distant tumors on a bilateral colorectal tumor model [24].

Other emerging combination strategies for improved RDT are gas therapy and PTT. Gas therapy releases CO molecules to cause mitochondria exhaustion and induce cell apoptosis [19]. The synergistic radiodynamic/gas therapy is bridged by RDT-produced OH^•^ that cleaves the coordination bond and releases CO. Fan et al. designed HMONs to be loaded with TBHP, a radiosensitizer, and Fe(CO)_5_, a CO-releasing molecule. As the irradiation dose increases, CO molecules released from HMOP-TBHP/Fe(CO)_5_ increase independent of oxygen concentration. Correspondingly, the treatment of HMOP-TBHP/Fe(CO)_5_ up-regulates caspase 3 and p53 and leads to more than 95% apoptosis and a higher rate of necrosis. TME not only brings resistance to cancer treatments but also initiates metastasis. PTT can alleviate tumor hypoxia by generating regional hyperthermia to ablate tumor cells and boost intratumoral blood flow that brings in oxygen [17]. Wang et al. reported that BiVO_4_@Bi_2_S_3_ HNRs can be converted into a photothermal agent under NIR irradiation. Depending on laser power density, they can demonstrate excellent thermal stability up to ~40 °C. The BiVO_4_@Bi_2_S_3_ HNR + NIR + X-ray treatment which employs both RDT and PTT leads to the lowest tumor cell survival rate without damaging surrounding tissues.

The synergistic antitumor treatment combining RDT with other therapies enhances cytotoxic ROS generation, mitigates the suppressive TME towards therapeutics, modulates immune responses and ultimately promotes overall efficacy. Along with the development of nanoscale drugs, future RDT administrations are expected to combine with various therapies for stability and biocompatibility.

## 6. Conclusions

This comprehensive review summarizes the recent progress in radiodynamic cancer therapy over the past two decades. Evolving from PDT, RDT distinguishes itself by leveraging X-rays and high-energy irradiation rays. The flexibility and loading capacity of nanomaterials have led to the development of responsive nanoscale PSs, addressing challenges such as limited tissue penetration, low energy transfer efficiency, and hypoxic TME. These PSs serve as a crucial link between RDT and other cancer therapies, including radiotherapy, gas therapy, and PTT, amplifying their therapeutic effects. RDT mechanisms involve the production of ROS upon low-dose irradiation, the direct excitation of high-efficiency PSs by low-dose X-rays, and the use of nanoparticles for targeted drug delivery to modulate the TME. Notably, ROS production, particularly ^1^O_2_, OH^•^, O_2_^•−^, and H_2_O_2_, proves highly cytotoxic to tumor cells, inducing damage to biomacromolecules and triggering cell cycle disruptions. Strategies employing composite nanomaterials, scintillators, and PSs for treating deep-seated tissues have shown promising potential, though further improvements in energy transfer efficiency and loading capacity are expected. Designing PSs that matches the excitation energy of low-dose X-rays, exploring nanoclusters as efficient and less toxic PSs, and developing biodegradable PSs are essential considerations for advancing RDT. Emerging combined strategies, including PTT, gas therapy, and checkpoint blockade immunotherapy, present innovative approaches to enhance RDT therapeutic outcomes. The ideal cancer therapy necessitates eradicating primary tumors, preventing metastasis, and overcoming the challenges posed by the TME. Multifunctional drugs responding to irradiation as PSs and inducing other cancer therapies hold promise as a developing strategy.

Treating deep-seated tumors with RDT, especially in combination with other modalities, showcases significant progress and promising perspectives. The ability to enhance radiotherapy, modulate immune responses, and synergize with gas and photothermal therapies positions RDT as a cornerstone in the evolving landscape of cancer treatment. Continued research and clinical exploration will unveil further breakthroughs, bringing us closer to more effective and targeted cancer therapies.

## Figures and Tables

**Figure 1 pharmaceutics-16-01135-f001:**
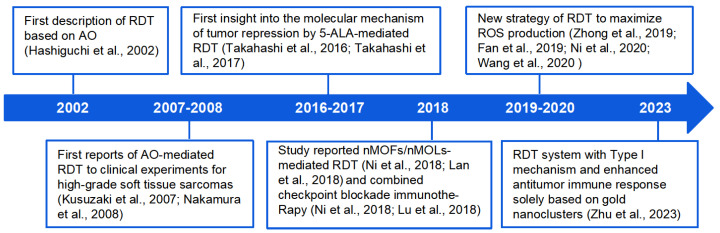
Historical timeline of some important developments regarding RDT [2,7,8,10,11,12,13,14,15,16,17,18,19].

**Figure 2 pharmaceutics-16-01135-f002:**
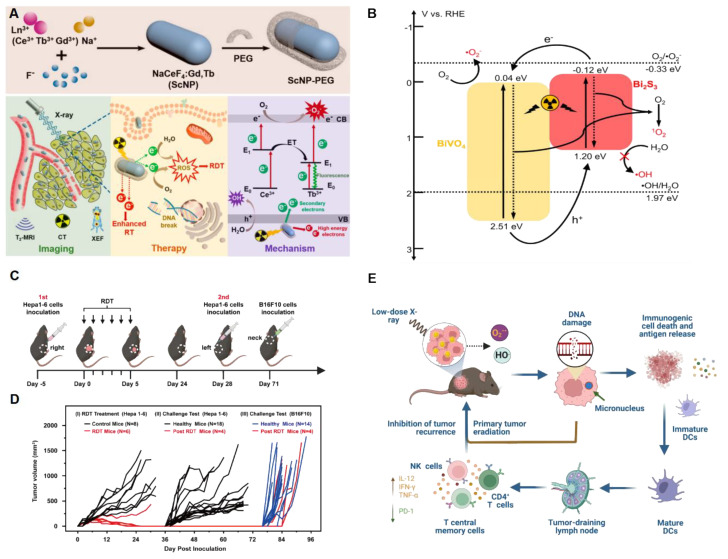
Schematic illustrations of RDT systems and/or mechanism. (**A**) X-ray-activated NaCeF4:Gd,Tb scintillator for XEF/CT/T2-MR imaging-guided synchronous RT and RDT of cancer. Reprinted with permission from reference [15]. Copyright 2019, American Chemical Society. (**B**) Illustration of energy band structure of BiVO_4_@Bi_2_S_3_ HNRs and energy transfer mechanism. Reprinted with permission from reference [17]. Copyright 2020, American Chemical Society. (**C**) Timeline design of animal experiments receiving RDT [18]. (**D**) Tumor growth curves in the tumor challenge study. The tumor size of each mouse was plotted separately in control (black or blue) and RDT group (red), showing that five out of six mice in RDT group were tumor free after treatment. The black and blue curves showed the tumor growth curves of three control groups with Hepa 1–6 (black) or B16F10 (blue) tumor cells injected on the same days. (**E**) AuNC@DHLA-mediated RDT and the corresponding mechanism under extremely low doses of X-rays. From reference [18]. CC BY 4.0.

**Figure 3 pharmaceutics-16-01135-f003:**
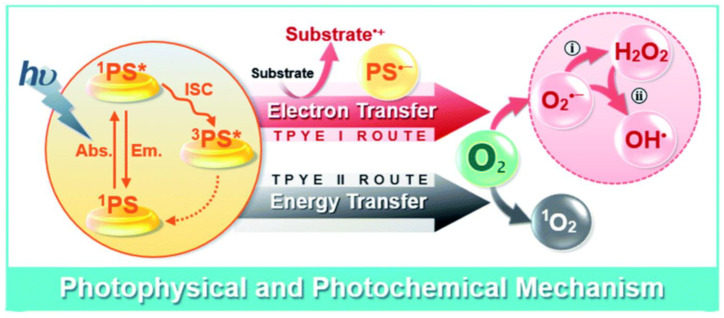
Schematic illustration of type I and type II photophysical and photochemical mechanisms. Asterisk (*) means excited state. Reprinted with permission from reference [44]. Copyright 2020, Royal Society of Chemistry.

**Figure 4 pharmaceutics-16-01135-f004:**
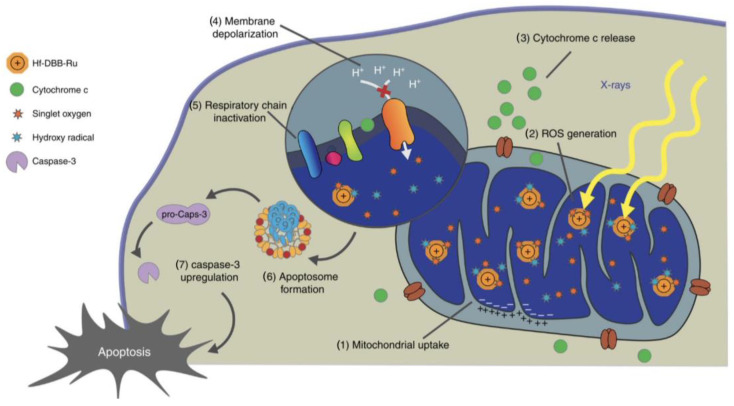
RT-RDT-induced apoptosis. Hf-DBB-Ru-targeting mitochondria were internalized by tumor cells and enriched in mitochondria owing to dispersed cationic charges in the nMOF. Under X-ray irradiation, the RT-RDT process triggered mitochondrial membrane potential depolarization, membrane integrity loss, respiratory chain inactivation, and cytochrome c release, initiating cancer cell apoptosis. From reference [7]. CC BY 4.0.

**Figure 5 pharmaceutics-16-01135-f005:**
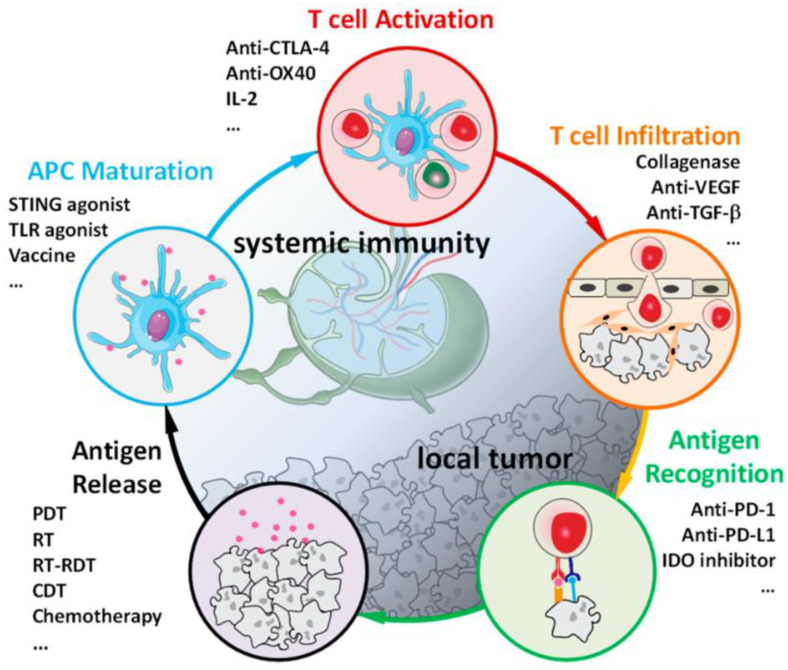
Immuno-oncology cycle and the use of immunotherapy to enhance key steps. This cycle starts with TA release from cancer cells, includes multiple steps of antigen presentation by mature antigen presenting cells, T cell activation in lymph nodes, and T cell trafficking and infiltration into tumor beds, and ends with the recognition of TAs by cytotoxic T lymphocytes (CTLs) to kill cancer cells. From reference [6]. CC BY 4.0.

**Figure 6 pharmaceutics-16-01135-f006:**
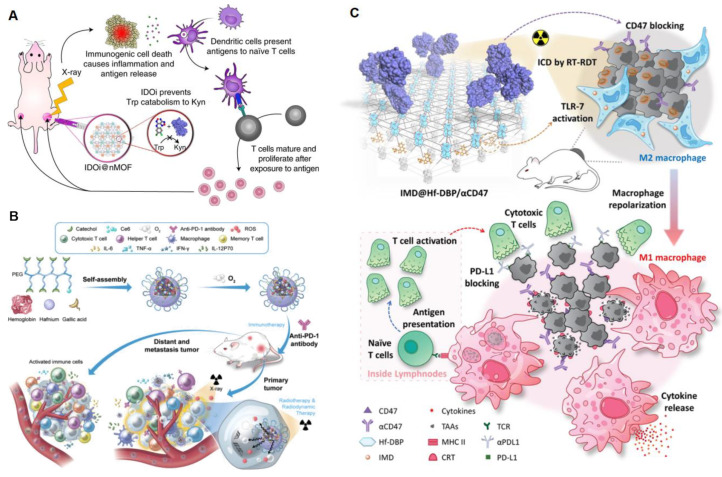
Schematic mechanism of combination of RDT and immunotherapy. (**A**) IDOi@nMOF overcame the suppressive TME by preventing Trp catabolism to Kyn and subsequent T cell anergy. Systemic IDOi activity combined with local RT–RDT induced immunogenic cell death, and antigen release led to the effective expansion and tumor infiltration of functional CD8+ T cells. Image from reference [13], reproduced with permission from SNCSC. (**B**) The co-administration of Hb@Hf-Ce6 and anti-PD-1 employs Hb as radiosensitizer to promote radiation dose enhancement and ROS production to inhibit the growth of the primary tumors and activates immune cells and immune cytokines to combat the distant and metastasis tumors. From reference [70]. CC BY 4.0. (**C**) IMD@Hf-DBP/αCD47 repolarizes M2 to M1 macrophages and promotes phagocytosis by blocking the “don’t-eat-me” signal on tumor cells. Reprinted with permission from reference [24]. Copyright 2020, American Chemical Society.

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
