# Peer review of "Treating Deep-Seated Tumors with Radiodynamic Therapy: Progress and Perspectives"

_pharmaceutics, 2024, doi:10.3390/pharmaceutics16091135_

Round 1

Reviewer 1 Report

Comments and Suggestions for Authors

The manuscript "Treating deep-seated tumors with radio-dynamic therapy: progress and perspectives" describes the advancement of radiation therapy with the use of photo- and radio-sensitizers. The authors provide some insights into mechanisms of radio-dynamic actions.
The text must be better organised in general and with language corrections. Please find specific comments below.

-(lines 17-19) Sentence "Challenges faced by the two-component strategy, including low loading capacity and inefficient energy transfer, hinder its final effectiveness" requires a more understandable formulation. It is not clear now if it is loading of cancer cells with particles or loading of particles with scintillating material.
-(lines 19-20) The next sentence "Challenges faced by the two-component strategy, including low loading capacity and inefficient energy transfer, hinder its final effectiveness" can also be a bit more specific Is it simply tumour hypoxia-mediated immunosuppression or something related to radiation.
-(lines 33-34) Another sentence "Surgeries carry risks of metastasis and recurrence due to large incisions" might be correct, however not understandable to other when one needs to relate large incisions to local and distant metastasis. The authors might find some keywords like "surgical stress" that could help to fix the sentence e.g. in this paper https://www.ncbi.nlm.nih.gov/pmc/articles/PMC5380551/
-(lines 40-42) One might agree that "co-localization of light, oxygen, and photosensitizers" is a logic way to describe the principle of PDT initiation, however it is better not to state that light is localised in this case and think of re-formulation of that sentence.
-(lines 42-44) "PDT is more advantageous because of the small wound it causes and its high selectivity and low side-effects". Small wound is debatable. High selectivity requires some number for tumour-helthy tissue ratio (if the authors have PS accumulation in mind). Alternativelly, some reference to some review paper can be used.
-(lines 47-54) That paragraph needs better wording. RDT advantages need references to publications. "higher excitation efficiency than PDT" needs clarification. Is x-rays more efficient in PS excitation (indirectly)? If the authors are backing up their statements on RDT later in the text, then they need to say it in that paragraph that those advantages are discussed further in the -folowing sections or something similar.
-(lines 78-79) Repetition of RDT descritpion. That sentence to be deleted or somehow supplemented.
-(line 84) "low-dose X-rays". Here the authors can be more specific and find in Hashiguchi et al. paper an exact dose.
-(lines 88-89) "lower local recurrence rate and increased specificity due to fluorescence"- not understandable how recurrence is decreased because of fluorescence directly
-(lines 91-92) "revealed corresponding molecular mechanism of tumor repression". What mechanism.
-(line 100) "have been applied to RDT". Applied and what are the results?
-(lines 100-102) "In principle, RDT can increase DNA double-strand break (DSB), halt cells at G2/M phase, enhance ROS generation, and induce immune responses that collectively result in the cell cycle disruption". Halt in G2 and disrupt. Not understandable. Please reformulate the sentence. ROS generation goes first.
-low-dose is repeated many times. What does it mean low. Relative to treatment dose? Is 50% of normal treatment dose low enough?
-(line 108) "low loading capacity, diffusion of photosensitizers". Loading of nanoparticles? Diffusion from nanoparticles, cells, tumour?
-References to figures in the text must be fixed.
-(line 151) "ROS that are more responsive to PDT"- bad formulation. "free radicals of ROS"- ROS is a subset of free radicals. That sentence asks for more work.
-(lines 200-201) "in downstream apoptotic pathways, resulting in dysfunction of mitochondria and apoptosis"- delete
-(line 205) "ROS generation destructs the DNA double strands". Please find an apropriate paper for ROS action on DNA. There is not only destruction of double strands. It is difficult to induce two breaks in close proximity with oxidation. Alternativelly you can call it a DNA damage.
-First paragraph of 4.2.2. It is not clear if double strand breaks are induced only by radiation itself or RDT has a big impact.
-(lines 240-241) "Apoptosis, necrosis, autophagy, and other cell death can serve as regulatory mechanisms of radio-dynamic therapy"- re-formulate
-(line 267) "Apoptosis and necrosis are merely induced under X-irradiation"- please provide a reference for this idea. It is not correct to say that apoptosis is not induced, however, it can be induced if more intensive radiation is given.
-(lines 443-444) "Gas therapy releases CO molecules to cause mitochondria and induce cell apoptosis [45]"- cause mitochondria is not clear
-In case of combination strategies, please indicate synergistic level in percent or other units
-If figures are taken from some publications, please make sure that it is reproduced with permission.
-The text could benefit very much from professional language correction services
-There is no information if other type of ionising radiation can induce radio-dynamic effect. Neutrons and accelerated protons or electrons can come into play.

Comments on the Quality of English Language

Corrections needed

Author Response

Dear Editor,

We would like to express our gratitude for your valuable feedback on our manuscript, titled "Treating deep-seated tumors with radio-dynamic therapy: progress and

Perspectives" (Manuscript ID: pharmaceutics-3098778), which we have submitted to Pharmaceutics.

The response to academic editor has been attached. We look forward to receiving your further guidance.

Sincerely,

Rongcheng Han, Ph.D.,

Associate Professor

Bio-Imaging Center,

State Key Laboratory of Molecular and Developmental Biology

Institute of Genetics and Developmental Biology, Chinese Academy of Sciences

Lincui East Road, Chaoyang District, Beijing 100101, China

Tel: 010-64806367;  E-mail: [email protected]

Response to Academic Editor

Comments from Academic Editor: Radio-dynamic therapy (RDT), as an emerging cancer treatment method, has attracted attention due to its remarkable therapeutic efficacy using low-dose, high-energy radiation (such as X-rays), and has shown significant potential in cancer treatment. The RDT system typically consists of scintillators and photosensitizers. Scintillators absorb X-rays and convert them to visible light, ac-tivating nearby photosensitizers to generate cytotoxic reactive oxygen species (ROS). Challenges faced by the two-component strategy, including low loading capacity and inefficient energy transfer, hinder its final effectiveness. In addition, the tumor microenvironment (TME) with hy-poxia and immunosuppression limits the efficacy of RDTs. Recent advances introduce one-component RDT systems based on nanomaterials with high-Z metal elements, which effectively inhibit deep-seated tumors. These novel RDT systems exhibit immune enhancement and immune memory, potentially eliminating both primary and metastatic tumors. This review comprehensively analyzes recent advances in the rational construction of RDTs, exploring their mechanisms and its application in the treatment of deep-seated tumors. Aimed at providing a practical resource for oncology researchers and practitioners, the review offers new perspectives for potential future di-rections in RDT research"

Response: We would like to express my gratitude to you for taking the time to review this manuscript and provide your professional opinion.

We have conducted a thorough review of the manuscript and implemented a series of comprehensive revisions and improvements. We hope that our efforts have met your expectations. We would like to express our gratitude once more for your recognition and encouragement.

We have carefully read through the entire manuscript and improved the text, language, etc.

  • All references have been substantially revised to be in accordance with the content of the manuscript.
  • All revisions to the manuscript are highlighted in red in Word's revision mode.

Reviewer 2 Report

Comments and Suggestions for Authors

Authors: Rongcheng Han, Shengcang Zhu, Siyue Lin 

Title: Treating deep-seated tumors with radio-dynamic therapy: progress and perspectives 

COMMENTS: 

This is a comprehensive review describing advances and challenges in radio-dynamic therapy of deep-seated tumors. This material wwould be interesting and helpful for radiotherapists and radiation oncologists. The submitted manuscript is well written and illustrated. The manuscript seems acceptable in the present form. However, I would recommend to view and discuss such a point as antioxidative potential/mechanisms of target tumor cells. Taking into consideration that ROS and O2 are the critical factors affecting the radiation response of target tumors, inhibitory pretreatment of certain antioxidative pathways/enzymes may sensitize deep-seated tumors to radio-dynamic therapy. I mean that reduced glutathione, thioredoxin, SOD, catalase and others may be targets for such radiosensitizing. What do you thinkk?     

Author Response

(The authors gave the same response as above.)
